# *Helicobacter pylori* and Epstein–Barr Virus Co-Infection in Gastric Disease: What Is the Correlation with *p53 Mutation*, Genes Methylation and Microsatellite Instability in a Cohort of Sicilian Population?

**DOI:** 10.3390/ijms24098104

**Published:** 2023-04-30

**Authors:** Anna Giammanco, Rita Anzalone, Nicola Serra, Giuseppa Graceffa, Salvatore Vieni, Nunzia Scibetta, Teresa Rea, Giuseppina Capra, Teresa Fasciana

**Affiliations:** 1Department of Health Promotion, Mother and Child Care, Internal Medicine and Medical Specialties, University of Palermo, 90127 Palermo, Italy; 2Department of Surgical Oncological and Oral Sciences, University of Palermo, 90133 Palermo, Italy; 3Department of Public Health, University Federico II of Naples, 80138 Napoli, Italy; 4Anatomopathology Unit, Arnas Civico Di Cristina Benfratelli Hospital, 90127 Palermo, Italy; 5Public Health Department, Federico II University Hospital, 80131 Naples, Italy

**Keywords:** mutation, microsatellite instability, *p53 mutation*, *H. pylori*, EBV

## Abstract

Genetic predisposition, environmental factors, and infectious agents interact in the development of gastric diseases. *Helicobacter pylori* (Hp) and Epstein–Barr virus (EBV) infection has recently been shown to be correlated with these diseases. A cross-sectional study was performed on 100 hospitalized Italian patients with and without gastric diseases. The patients were stratified into four groups. Significant methylation status differences among CDH1, DAPK, COX2, hMLH1 and CDKN2A were observed for coinfected (Hp-EBV group) patients; particularly, a significant presence of COX2 (*p* = 0.0179) was observed. For microsatellite instability, minor stability was described in the Hp-HBV group (69.23%, *p* = 0.0456). Finally, for *p53 mutation* in the EBV group, exon 6 was, significantly, most frequent in comparison to others (*p* = 0.0124), and in the Hp-EBV group exon 8 was, significantly, most frequent in comparison to others (*p* < 0.0001). A significant positive relationship was found between patients with infection (Hp, EBV or both) and *p53 mutation* (rho = 0.383, *p* = 0.0001), methylation status (rho = 0.432, *p* < 0.0001) and microsatellite instability (rho = 0.285, *p* = 0.004). Finally, we observed among infection and methylation status, microsatellite instability, and *p53 mutation* a significant positive relationship only between infection and methylation status (OR = 3.78, *p* = 0.0075) and infection and *p53 mutation* (OR = 6.21, *p* = 0.0082). According to our analysis, gastric disease in the Sicilian population has different pathways depending on the presence of various factors, including infectious agents such as Hp and EBV and genetic factors of the subject.

## 1. Introduction

Genetic predisposition, environmental factors and infectious agents interact in the development of gastric diseases. Hp and EBV infection has recently been shown to be associated with the development of gastric diseases. Gastric carcinoma (GC) remains a common disease with a dismal prognosis. Gastric carcinogenesis is a multistep process accompanied by the accumulation of alterations in critical regulatory genes. In particular, the development of genome-wide analysis tools has enabled the discovery of genetic and epigenetic modifications in gastric cancer; for example, aberrant DNA methylation in gene promoter regions is supposed to play a critical position in gastric carcinogenesis [1,2,3].

Hp infection and dietary factors are significant environmental risks for GC. Numerous genetic variations have been related to the tumorigenesis of GC [4,5]. These modifications contain amplification of oncogenes, mutations in tumor-suppressor genes and the dysfunction of mismatch repair genes. Changeability in the biological selves of GC may be linked to these genetic variations; therefore, these intricate genes may reproduce the collection of their causes and diverse histological subtypes. The etiological viewpoint is also essential for the study of gastric cancers, and two distinct pathogens, Hp and EBV, are known to participate in gastric carcinogenesis [6].

After two decades of research, the role of Hp in certain types of gastric diseases is extensively accepted, comprising bacterium eradication as part of its cure [7,8]. Hp is a spiral-shaped bacterium that grows in the mucus layer of the human stomach, causing inflammation known as gastritis. Additionally, the infection is correlated with ulcers, long-term anemia, gastritis, and gastric cancer [9,10,11,12]. Hp is largely spread through polluted food, food, saliva, or mouth-to-mouth contact and probably transmitted sexually via oral–genital contact [13]. Nearly 50% of the worldwide population is estimated to be infected by this bacterium [14], in less than 2% of which it is responsible for gastric cancer [15]. This microorganism is acquired during childhood in all countries [16] and frequently in developing countries. Moreover, the infection percentage of children in developing countries is higher than that in developed countries, 80% compared to 10% at the age of 20 years, while senior individuals in both types of nations have about 50% rates of infection at 60 years of age [17].

The virus EBV belongs to a gamma DNA herpes virus that infects over 95% of the global population and remains as a symptomless life-long infection [18]. The main targets of this virus are the B lymphocytes and epithelial cells. EBV was found to be related to an enormous number of human diseases, such as lymphomas, nasopharyngeal carcinoma (NPC) and gastric carcinoma. Gene methylation following EBV infection might be a response of the host cell against foreign DNA; on the other hand, it might benefit EBV by allowing it to seep into the immune response of the host [19]. Epigenetic deviations in tumorigenesis produce DNA methylation and indicate maximum invasiveness.

DNA hypermethylation is the most significant hallmark of EBV-linked gastric cancer that discriminates it from other molecular subtypes of gastric cancer [20]. EBV-positive tumors show a distinctive promoter hypermethylation outline that reflects their spectrum of mutations and gene expression [21]. Of specific attention, all EBV-positive tumors display CDKN2A but not MLH1 promoter hypermethylation, which characterizes the microsatellite instability subtype of gastric cancer [22].

Many studies have acknowledged transcriptional silencing by DNA methylation as a mechanism accountable for tumor suppressor inactivation. Methylation of promoter CpG islands leads to DNA structural changes and, consequentially, gene deactivation [23]. Various cancers, comprising gastric tumors, demonstrate methylation of multiple genes plus CDH1, DAPK, COX2, hMLH1 and CDKN2A [24].

Microsatellite instability reflects a wrong form of DNA replication in repetitive microsatellite sequences and has been a sensitive hallmark of mismatch repair gene inactivation. Microsatellite instability has been related with less aggressive tumor behavior and auspicious prediction in infrequent colorectal cancer [25]. Microsatellite instability status has been evaluated by means of BAT26 mononucleotide repeats because this indicator is quasi-monomorphic in normal DNA and has exposed high sensitivity and specificity in the identification of the microsatellite instability phenotype [26].

The main alterations in the process of gastric carcinogenesis include the mutation of the p53 gene. This mutation has also been described in pre-malignant lesions of the stomach, such as chronic gastritis, intestinal metaplasia and dyspepsia [27]. Kodama et al. (2007) suggested an accretion of wild-type p53, particularly in Hp infected mucosa, possibly due to Hp-induced DNA damage [28]. In our study, we report the highest frequency of *p53 mutation* in the groups with Hp infection, especially among the *cagA* positive cases, supporting previous studies [29]. The raised percentage of *p53 mutation* in EBV-associated and EBV-negative gastric carcinomas was noticed by Lima et al. (2008), demonstrating that the *p53 mutation* is a linked modification in the infection self-regulation of gastric carcinogenesis [30].

This evaluation is important because, for example, early gastric cancer has peculiar molecular characteristics compared to advanced stages [31,32]. Diffuse-type and intestinal-type gastric cancer (the second linked to Hp infection) also have different molecular characteristics as demonstrated by The Cancer Genome Atlas (TCGA) [33]. TCGA also shows that methylation is characteristic of two gastric cancer subtypes (EBV and MSI) in different genes. CDH1 mutation and/or methylation is a common phenomenon in diffuse gastric cancer (more linked to genetic factors and less to environmental factors), but some authors reported that alterations in the methylation pattern are present also in the intestinal histotype and that Hp is linked to CDH1 methylation [34,35].

The purpose of our study was to evaluate the correlations between infection such as Hp, EBV or both and *p53 mutation*s, DNA methylation and microsatellite instability in different gastric diseases in people living in the Mediterranean area. The search was considered stimulating since Sicily, our geographic location of choice, constitutes one of the most complex mixtures of diverse ethnic elements in Europe.

## 2. Results

The defined groups for this study, according to infection type, are reported as following:The group non-infected (NIG) comprised 45 patients without infection, with 44.44% males and 55.56% females, with ages in the range 22–87, with a mean of 57.67 y.o. and standard deviation (SD) of 18.02 y.o.;The group Hp comprised 18 patients with Hp infection only, with 27.78% males and 72.22% females, with ages in the range 31–80, with a mean of 58.53 y.o. and a standard deviation (SD) of 15.26 y.o.;The group EBV comprised 11 patients with EBV infection only, with 45.45% males and 54.55% females, with ages in the range 25–77, with a mean of 61.09 y.o. and a standard deviation (SD) of 14.31 y.o.;The group EBV-Hp comprised 26 patients with co-infection by EBV and Hp, with 26.92% males and 73.08% females, with ages in the range 20–87, with a mean of 58.58 y.o. and a standard deviation (SD) of 16.23 y.o.

In Table 1, we report the clinical information of the total patient sample, including age, gender, symptoms and infection type, with patients infected by Hp, EBV or both (coinfected).

Table 2 shows clinical information such as age, gender, symptoms, methylation status, microsatellite instability and *p53 mutation* among patients without infection, with Hp infection only, with EBV infection only and patients with both Hp and EBV infection.

Regarding analysis within groups, from Table 3, we observe, for gender, no significant differences for the no-infected group (*p* = 0.46), Hp group (*p* = 0.06) and EBV group (*p* = 0.76), while for the Hp-EBV group, a significant difference was observed (M: 26.92% vs. F: 73.08%, *p* = 0.0186).

For symptoms, no significant differences among patients with normal gastric mucosa, active chronic gastritis, gastric cancer and MALT lymphoma were observed in the no-infected group (*p* = 0.85), Hp group (*p* = 0.49), EBV group (*p* = 0.63) and Hp-EBV group (*p* = 0.57).

For methylation status, no significant differences among CDH1, DAPK, COX2, hMLH1 and CDKN2A were observed for the no-infected group (*p* = 0.86), Hp group (*p* = 0.86) and EBV group (*p* = 0.14), while for coinfected (Hp-EBV group) patients, we observed a significant presence of COX2 (*p* = 0.0179).

For microsatellite instability, for each group, we observed a significantly low presence of microsatellite instability (NIG: 4.44%, Hp: 27.78%, EBV: 9.09%, Hp-EBV = 30.77%).

Finally, for *p53 mutation*, significant differences among exon 5, exon 6, exon 7, exon 8 and exon 9 were only found in the Hp-EBV group, where exon 8 was significant most frequently in comparison to others (*p* < 0.0001).

Regarding analysis among groups, from Table 3, we observe a significant test for microsatellite instability; in particular, a minor stability was individualized in the Hp-HBV group (69.23%, *p* = 0.0458) and for *p53 mutation*, where exon 8 was significantly less frequent in the NIG group and exon 7 (*p* = 0.0298) in the Hp group (*p* = 0.0415).

Table 4 shows the relationship analysis between infection and age, gender, symptoms, methylation status, microsatellite instability and *p53 mutation*.

From Table 4, a significant relationship between patients with infection (Hp, EBV or both) and *p53 mutation* (*p* = 0.0001), methylation status (*p* < 0.0001) and microsatellite instability (*p* = 0.0052) was observed. In other words, the presence of infection is correlated with *p53 mutation* or the presence of methylation status or microsatellite instability.

In Table 3, we perform the logistic regression between the infection variable and significant predictors defined in Table 5.

From logistic regression, we found that among methylation status, microsatellite instability and *p53 mutation*, there was a significant positive relationship only between infection and methylation status (OR = 3.78, *p* = 0.0075) and infection and *p53 mutation* (OR = 6.21, *p* = 0.0082); this shows that methylation status and *p53 mutation* are more correlated to infection than microsatellite instability (OR = 3.10, *p* = 0.19).

## 3. Discussion

In this study, we evaluated the relationship between Hp and EBV infection with *p53 mutation*, methylation status and microsatellite instability in gastric diseases in people living in the Mediterranean area [36,37,38,39] using PCR and in situ hybridization.

Our results show that Hp and EBV infection were present in 18% and 11% of the patients while coinfection was found in 26% of patients. Particularly, in patients with Hp infection, 27.78% were males and 72.22% were females, while in patients with EBV infection, 45.45% was observed in males and 54.55% in females. Finally, in patients with coinfection, 26.92% were males and 73.08% were females.

In previous studies, the highest rates of Hp and EBV infection were found in the male gender, in contrast to our results; this is probably due to the high presence of females in our sample [64%], and it is probable that possible environmental and epigenetic factors make the female sex more susceptible [40,41].

In our study, we considered a patient sample with different gastric symptoms such as active chronic gastritis, gastric cancer and MALT lymphoma, also including patients without symptoms, i.e., patients with normal gastric mucosa.

Hp was included among carcinogen agents as a Class 1 carcinogen in 1994 [42]. Patients with chronic Hp infection have physiological and morphological changes within the gastric environment and are at risk for neoplastic transformation. Recently, it was established that EBV is also connected to the development of gastric carcinoma. The association between EBV infection and gastric cancer has been confirmed by strong pieces of evidence such as the monoclonality of the viral genome and its presence in almost all tumor cells [43]. Moreover, there is sufficient indication to propose that genetic aspects can contribute to the exhibition of efficient gastrointestinal disorders. For example, polymorphisms in genes that code cytokines affect cytokine secretion levels and appear to contribute to the risk of gastric diseases [39,44].

In addition, Hp and EBV co-infection may be central to the aberrant expression of p53 protein, complete with host DNA damage, and the methylation of multiple genes including CDH1, DAPK, COX2, hMLH1 and CDKN2A can lead to DNA structural changes and gene deactivation.

Finally, MSI has been painstakingly evidenced to be a hallmark of mismatch repair gene inactivation [45].

In our study, the positivity rate of Hp in patients with normal gastric mucosa was higher than in patients with gastric cancer, active gastritis chronic and MALT lymphoma, at rates of 38.89%, 16.67%, 16.67% and 27.78%, respectively.

In patients with EBV infection, in 36.36% of the patients, GC was observed, while in patients with normal gastric mucosa, active gastritis chronic and Malt lymphoma, it was observed in 9.09%, 27.27% and 27.27%, respectively. The high percentage of EBV infection in patients with gastric cancer suggests that EBV plays a crucial role in tumorigenesis, approaching the close correlation of EBV with nasopharyngeal lymphoepithelioma [20,46,47,48]. Our data suggest that EBV may have a role in the development of GC in Hp negative patients; the mechanism of a minor rate of Hp in negative patients may be diverse and reflect the genetic vulnerability of the infections or the gastric milieu in EBV-associated GC being unable to support Hp.

In any case, the rate of Hp in all patients analyzed was 44% and the rate of EBV was 37%, including coinfected patients.

In patients with KG, the frequency of coinfection was higher than in those with normal gastric mucosa, active chronic gastritis, and MALT lymphoma, at 34.62%, 15.38%, 26.92% and 23.08%, respectively. In any case, the frequency of Hp and EBV infection found in this study agrees with reported results from various world regions.

Despite the recognized condition of Hp and EBV in the gastric cancer etiology, insufficient studies have acknowledged the interrelation of these two agents in gastric cancer cases. Thus, we investigated the presence of both Hp and EBV, in parallel with the status of DNA methylation, microsatellite instability and the mutation of tumor suppressor p53.

We assumed that these two microorganisms may be affected by each other or may play significant roles together directly or indirectly in the pathogenesis of gastric disease [46].

For methylation status, in CDH1, DAPK, COX2, hMLH1 and CDKN2A, no important alterations were detected in the no-infected group (*p* = 0.86), Hp group [*p* = 0.86] and EBV group (*p* = 0.14), while in the coinfected group (Hp-EBV), we observed a significant presence of COX2 (*p* = 0.0179). The relationship between COX2 methylation and gene downregulation has been well recognized in the literature [47]. COX2 overexpression is related with enhanced proliferation, angiogenesis, resistance to apoptosis and tumorigenesis [48]. Despite the deceptive choosy benefit given by COX2 overexpression, the results from our investigate group and others [20] propose that COX2 overexpression may not be important in all cases of gastric tumorigenesis [49].

To underline the participation of two pathogens in the expansion of gastric disease, our data show that in co-infected patients, we observe a higher microsatellite instability (*p* = 0.0121) and a lower microsatellite stability (69.23%, *p* = 0.0456) respective to other groups by means of post hoc chi-square.

Finally, the present study reports the highest incidence of *p53 mutation* in the patients with co-infection, and in particular, exon 8 presents a high rate of mutation. These data are supported by Eeles et al. 1993, who reported the correlation with exon 8 mutation and development of multiple independent benign and malignant tumors [50].

The mutation of the p53 gene was relevant in all groups, probably representing that it was not only correlated with the infection agent.

In the EBV group, exon 6 of the p53 gene was, significantly, most frequent in comparison to the others (*p* = 0.0124), and in the Hp-EBV group, exon 8 was, significantly, most frequent in comparison to the others (*p* < 0.0001).

Among the variables analyzed, the relationship analysis shows an association between the infection status, *p53 mutation*, the methylation status and the presence of microsatellite instability; in other words, the presence of infection is correlated with the mutation of p53 and the presence of methylation status or microsatellite instability.

In addition, via logistic regression between infection and significant predictors such as mutation of p53 and the presence of methylation status and microsatellite instability, this resulted in a significant positive relationship only between infection and methylation status (OR = 3.78, *p* = 0.0075) and infection and *p53 mutation* (OR = 6.21, *p* = 0.0082); this shows that methylation status and *p53 mutation* were more correlated to infection than microsatellite instability (OR = 3.10, *p* = 0.19).

The significant positive relationship that resulted between infection and *p53 mutation* confirms that methylation is an premature epigenetic incident in the molecular alteration of gastric disease.

The pattern of methylation in the genes analyzed proposes that gastric disease can happen by diverse pathways according to different environmental and epigenetic factors.

A significant positive relationship only resulted between infection and methylation status and infection and *p53 mutation*, which shows that genomic structural integrity in the development of gastric disease is important.

The correlation between MSI and clinical characteristics of GC remains unknown. However, some research has described MSI gastric tumors as linked with different tumor locations, isotypes, less metastases and good prognosis [51].

As far as we understand, the present study is one of the principal studies of the molecular epidemiology of infectious carcinogenic microorganisms in gastric samples from Southern Italy.

## 4. Materials and Methods

### 4.1. Patients and Sample Collection

A cross-sectional study was performed on a sample of 100 patients, 37% males and 63% females, with ages in the range 20–87, mean 58.43 y.o. and standard deviation (SD) 16.52 y.o.

The DNA was extracted from biopsy sampling in healthy, inflammatory and tumor mucosa. We stratified the samples according to infection type and obtained four groups: non-infected, Hp infected, EBV infected and co-infected (Hp + EBV). The groups were defined by a total of 100 patients enrolled in a random way and comprised 25 patients with normal gastric mucosa, 25 patients with active chronic gastritis, 25 patients with gastric cancer and 25 patients with MALT lymphoma who had Hp infection, EBV, or both. The four groups obtained according to infection type were defined until 100 patients were reached.

### 4.2. Exclusion Criteria

All patients excluded from the study were as follows: previous attempts to eradicate Hp and the use of antibiotics or proton pump inhibitors within two weeks prior to endoscopy.

### 4.3. Histology Analysis

The diagnosis of gastroduodenal disease was based on endoscopic and histological examination, and it was established according to the Sidney System classification [36]. DNA was isolated from gastric biopsies, and Hp and EBV DNA were detected via PCR methodology. The genomic DNA of the biopsies was extracted using a High Pure Template Preparation kit [Roche], in accordance with the manufacturer’s instructions. The extracted DNA was stored at −20 °C until use. Hp infection was diagnosed by the detection of *ureaseA* gene using nested PCR, while the BAMHI-W fragment region of the EBV genome was used as the target to evaluate the presence of the virus, according to Di Carlo et al. (2011) and Giardina et al. (2008) [37,38].

### 4.4. Valuation of p53 Mutation and Single-Stranded Conformation Polymorphism [SSCP] Analysis

Primers P1 and P2 (Table 5) were used in a standard PCR to amplify the 2.9 kb fragment of the genomic DNA containing exons 5 to 9 of the p53 gene [39]. The amplified fragments were purified from a 1% agarose gel after electrophoresis. For the SSCP analyses, PCR was performed as described by Effert et al. (1996) [52]. Distinct primer pairs were used to amplify exons 5 to 8 of the p53 gene in separate PCRs for 30 cycles at 94, 55 and 72 °C. Five microliters of the PCR product was used, and SSCP analysis was performed as described by Effert et al. (1996) [52].

In Table 1, we report the sequences of oligonucleotides used for the evaluation of *p53 mutation*s.

### 4.5. Bisulfite Modification and Methylation-Specific PCR [MSP]

DNA from the tissues was subjected to treatment with sodium bisulfite as described by Herman at al. (1996) [53]. The modified DNA was amplified with primers specific for either the methylated or unmethylated sequences of hMLH1, COX2, DAPK, CDKN2A and CDH1 [Table 2]. PCR was individually performed as described by Ferrasi et al. (2010) [54]. In Table 6, we report the primer sequences and PCR conditions for methylation-specific PCR [MSP] analysis.

### 4.6. Microsatellite Instability Analysis

Microsatellite instability analysis was performed using the BAT26 primer set [5’-TGACTACTTTTGACTTCAGCC-3’ sense and 5’-AACCATTCAACATTTTTAACCC-3’ antisense]. The sense primer was labeled with 6-FAM. PCR was performed in a final volume of 25 μL containing 1 × PCR buffer, 3.0 mmol/L MgCl2, 0.2 μmol/L dNTPs, 0.4 μmol/L of each primer, 2 U of Platinum Taq DNA Polymerase (Invitrogen, Waltham, MA, USA) and 50 ng of DNA. The thermal conditions were 94 °C/5 min followed by 40 cycles [94 °C/1 min, 50 °C/1 min and 72 °C/1 min] and a final extension at 72 °C/7 min. The dye-labeled PCR products were analyzed with a ABI PRISM 3130 Genetic Analyzer using Genescan 3.7 software (Applied Biosystems, Waltham, MA, USA) according to Hoang JM et al. (1997) [36,55].

After electrophoresis, gels were dried at 80 °C and exposed to radiograph film. The band pattern was compared between tumorous and non-tumorous tissues for each patient. To avoid PCR artifacts, all positive tests were duplicated. Only cases with microsatellite alterations at 3 or more loci [≥30% frequency], only in neoplastic tissue, were ascribed to microsatellite instability.

### 4.7. Ethical Considerations

Informed consent was signed by all participants in the study. Anonymity was guaranteed for all participants. No economic incentives were offered or provided for participation in this study. The study was performed under the ethical considerations of the Helsinki Declaration. The study was approved by the Local Ethics Committee.

### 4.8. Statistical Analysis

Data are presented as number and percentage for categorical variables and continuous data are expressed as mean ± standard deviation [SD], unless otherwise specified. A binomial test was performed to compare two mutually exclusive proportions or percentages in the groups. A multi-comparison chi-square test was used to define significant differences among groups; if the chi-square test was positive [*p*-value < 0.05], then residual analysis with continuity correction for the Z-test was performed to localize the highest or lowest significant presence. In addition, for the analyses among three or more modalities of a variable, the chi-square goodness of fit was used. The test for normal distribution was performed using a Shapiro–Wilk test. For samples not normally distributed, the Kruskal–Wallis test was performed in multi-comparison among three or more unpaired samples, and if the Kruskal–Wallis test was significant (*p*-value < 0.05), a post hoc test using the Conover test for pairwise comparison of subgroups was performed.

The relationship between infections and other parameters was calculated using a chi square test or Fisher’s exact test (dichotomous vs. dichotomous) or a Mann–Whitney test (dichotomous vs. no normal continuous data). For this step, we define the following variables:Infections: yes (including patients with Hp, EBV, or both) = 1, no = 0;Gender: male = 1, female = 0;Symptoms: yes (including active chronic gastritis, gastric cancer and MALT lymphoma) = 1; no (normal gastric mucosa) = 0;*p53 mutation*: yes (if among exon 5–9 there was a mutation) = 1, no = 0;Methylation status: yes (if among the genes CDH1, DAPK, COX2, hMLH1, CDKN2A there was a methylation) = 1, no = 0;Microsatellite instability: yes = 1, no = 0.

Logistic regression was used to find the best-fitting model to describe the relationship between the dichotomous characteristic of interest (dependent variable) and a set of independent variables.

All tests with p < 0.05 were considered significant. All data were analyzed with Matlab statistical toolbox version 2008 (MathWorks, Natick, MA, USA) for 32-bit Windows.

## 5. Conclusions

Our preliminary research data will encourage the development of large-series case–control study models based on new or developed methods that will be aimed at demonstrating the molecularly based physiological possible synergistic crosstalk relationship of Hp and EBV inside and outside the gastric epithelium and their mutual interactions with host immune responses. As suggested by Sander at al. (2014), our paper underlines the necessity to develop a personalized medicine with the aim to evaluate all aspects involved in specific diseases and explore therapies in defined sets of patients, ultimately improving survival from this deadly disease [56]. 

## 6. Limitations

Some statistical analyses were performed on small numbers of sample data; in addition, in the regression analysis, we formalized some variables as dichotomous, increasing the probability of statistical bias. To reduce the presence of statistical bias, in the first case we used the statistical test for small samples and, where necessary, continuity corrections. In the second case, we observed that the use of dichotomous variables was adequate with our objective, i.e., to investigate only the impact of variables such as the presence of *p53 mutation*, methylation status or microsatellite instability, without considering the type of infection, *p53 mutation*, methylation status, or microsatellite instability. A multicenter study with a very large sample will be the next step of this study to confirm these preliminary results.

## Figures and Tables

**Table 1 ijms-24-08104-t001:** Clinical and genetic information: age, gender, symptoms and infection type in total patient sample.

Parameters	Sample Data
Patients	100
Age	58.43 ± 16.52 *
Gender	
Male	37%
Female	64%
Symptoms	
NGM	25%
GCA	25%
GC	25%
ML	25%
Analysis with PCR	
No infected	45%
Infected by Hp	18%
Infected by EBV	11%
Co-infected	26%
*p53 mutation*	
exon 5	2%
exon 6	8%
exon 7	1%
exon 8	13%
exon 9	2%

* The age reported is the age at sampling. NGM: normal gastric mucosa; GCA: active chronic gastritis; GC: gastric cancer; ML: MALT lymphoma; Hp: *H. pylori*; coinfected: patients with Hp and EBV; EBV: Epstein–Barr virus.

**Table 2 ijms-24-08104-t002:** Characteristics of the groups: no-infected, patients with Hp, with EBV and with both.

Parameters	No-Infected Group (*n* = 45)	Hp Group (*n* = 18)	EBV Group (*n* = 11)	Hp-EBV Group *(n* = 26)	Statistical Analysis among Groups *p*-Value (Test)
Age					*p* = 0.97 (KW)
Mean ± SD	57.67 ± 18.02	58.53 ± 15.26	61.09 ± 14.31	58.58 ± 16.23	
Median (IQR)	62 (42.75, 72)	63 (41.75, 69.5)	64 (58, 68.5)	61.5 (52, 68)	
Gender					
Male	44.44%[20]	27.78%[5]	45.45%[5]	26.92%[7]	*p* = 0.36 (F)
Female	55.56%[25]	72.22%[13]	54.55%[6]	73.08%[19]	
	*p* = 0.46 (B)	*p* = 0.06 (B)	*p* = 0.76 (B)	*p* = 0.0186 (B) *	
Symptoms					
NGM	52%[13]	38.89%[7]	9.09%[1]	15.38%[4]	*p* = 0.66 (F)
GCA	48%[12]	16.67%[3]	27.27%[3]	26.92%[7]	
GC	36%[9]	16.67%[3]	36.36%[4]	34.62%[9]	
ML	44%[11]	27.78%[5]	27.27%[3]	23.08%[6]	
	*p* = 0.85 (C)	*p* = 0.49 (C)	*p* = 0.63 (C)	*p* = 0.57 (C)	
Methylation status					
CDH1	8.89%[4]	16.67%[3]	18.18%[2]	19.23%[5]	*p* = 0.61 (F)
DAPK	8.89%[4]	16.67%[3]	0.00%[0]	7.69%[2]	
COX2	6.67%[3]	16.67%[3]	36.36%[4]	46.15%[12]	
hMLH1	4.44%[2]	11.11%[2]	18.18%[2]	15.38%[4]	
CDKN2A	4.44%[2]	5.56%[1]	0.00%[0]	23.08%[6]	
	*p* = 0.86 (C)	*p* = 0.25 (C)	Test no realiable	*p* = 0.044 * (C) COX2 **, *p* = 0.0179 * (Z)	
Microsatellite instability					
MSS	95.56%[43]	72.22%[13]	90.91%[10]	69.23%[18]	*p* = 0.0069 * (F) MSS (Hp-EBV) ***, *p* = 0.0458 (Z)
MSI	4.44%[2]	27.78%[5]	9.09%[1]	30.77%[8]	
	*p* < 0.0001 * (B)	*p* = 0.059 (B)	*p* = 0.0067 * (B)	*p* = 0.0499 * (B)	
*p53 mutation*					
exon 5	2.22%[1]	0.00%[0]	0.00%[0]	3.85%[1]	*p* = 0.024 * (F) exon 8 (NIG) ***, *p* = 0.0298 (Z) exon 7 (Hp) ***, *p* = 0.0415 (Z)
exon 6	4.44%[2]	5.56%[1]	36.36%[4]	3.85%[1]	
exon 7	0.0%[0]	0.00%[0]	0.00%[0]	3.85%[1]	
exon 8	0.0%[0]	5.56%[1]	9.09%[1]	42.31%[11]	
exon 9	0.0%[0]	0.00%[0]	0.00%[0]	7.69%[2]	
	Test not realiable	Test no realiable	Test no realiable	*p* < 0.0001 * (C) exon 8 **, *p* < 0.0001 * (Z)	

*: significant test; B: binomial test; C: chi-square test; Z: z test; **: significantly more frequent; ***: significantly less frequent. F: Fisher’s exact test; KW: Kruskal–Wallis test; NGM: normal gastric mucosa; GCA: active chronic gastritis; GC: gastric cancer; ML: MALT lymphoma; Hp: *H. pylori*; coinfected: patients with Hp and EBV; EBV: Epstein–Barr virus.

**Table 3 ijms-24-08104-t003:** Logistic regression analysis between infection and significant factors described in Table 4.

Logistic Regression	Coefficient	Standard Error	OR	95% CI	*p*-Value
Null model vs. full model					<0.0001 [C]
Infection/*p53 mutation*	1.83	0.69	6.21	1.6; 24.07	0.0082 *
Infection/methylation status	1.33	0.50	3.78	1.43; 10.01	0.0075 *
Infections/microsatellite instability	1.13	0.85	3.10	0.58–16.55	0.19
Constant	–1.03	0.35			0.0034 *

*: significant test; OR: odds ratios; CI: odds ratios confidence interval at 95%; the null model: −2ln [L0], where L0 is the likelihood of obtaining the observations if the independent variables do not affect the outcome; the full model: −2ln [L0], where L0 is the likelihood of obtaining the observations with all independent variables incorporated in the model; C: chi-square test.

**Table 4 ijms-24-08104-t004:** Relationship analysis between infection and age, gender, symptoms, methylation status, microsatellite instability and *p53 mutation*.

Parameters	Infected	No Infected	*p*-Value (Test)
Infection/age	59.1 ± 15.3 63 (48, 69)	57.7 ± 18.0 62 (42.75, 72)	0.85 (MW)
Infection/gender	38F, 17M	25F, 20M	0.17 (C)
Infection/symptoms	12 (no), 43 (yes)	13 (no), 32 (yes)	0.42 (C)
Infection/*p53 mutation*	33 (no), 22 (yes)	42 (no), 3 (yes)	0.0001 * (F)
Infection/methylation status	13 (no), 42 (yes)	30 (no), 15 (yes)	<0.0001 * (C)
Infections/microsatellite instability	41 (no), 14 (yes)	43 (no), 2 (yes)	0.0052 * (F)

*: significant test [*p* < 0.05]. C: chi square test, F: Fisher’s exact test, MW: Mann–Whitney test; no: no presence; yes: presence.

**Table 5 ijms-24-08104-t005:** Sequences of oligonucleotides used for evaluation of *p53 mutation*.

Exon	Primer	Sequence	Fragment Length
	P1	GACGGAATTCGTCCCAAGCAATGGATGAT	2.9 kb
	P2	GTCAGTCGACCTTAGTACCTGAAGGGTGA	
5	P3	TTCCTCTTCCTGCAGTACT	209 bp
	P4	AGCTGCTCACCATCGCTAT	
6	P5	GGCCTCTGATTCCTCACTGA	170 bp
	P6	GCCACTGACAACCACCCTTA	
7	P7	TGTTGTCTCCTAGGTTGGCT	139 bp
	P8	CAAGTGGCTCCTGACCTGGA	
8	P9	CCTATCCTGAGTAGTGGTAA	164 bp
	P10	TCCTGCTTGCTTACCTCGCT	
9	P9	CCTATCCTGAGTAGTGGTAA	320 bp
	P2	GTCAGTCGACCTTAGTACCTGAAGGGTGA	

**Table 6 ijms-24-08104-t006:** Primer sequences and PCR conditions for methylation-specific PCR [MSP] analysis.

Gene	Primer [5′-3′] Forward	Primer [5′-3′] Reverse	Size [bp]
COX2	M TTAGATACGGCGGCGGCGGC	TCTTTACCCGAACGCTTCCG	161
	U ATAGATTAGATATGGTGGTGGTGGT	CACAATCTTTACCCAAACACTTCCA	171
DAPK	M GGATAGTCGGATCGAGTTAACGTC	CCCTCCCAAACGCCGA	98
	U GGAGGATAGTTGGATTGAGTTAATGTT	CAAATCCCTCCCAAACACCAA	116
CDH1	M TTAGGTTAGAGGGTTATCGCGT	TAACTAAAAATTCACCTACCGAC	115
	U TAATTTTAGGTTAGAGGGTTATTGT	CACAACCAATCAACAACACA	97
hMLH1	M TATATCGTTCGTAGTATTCGTGT	ACCACCTCATCATAACTACCCACA	153
	U TTTTGATGTAGATGTTTTATTAGGGTTGT	ACCACCTCATCATAACTACCCACA	124
CDKN2A	M TTATTAGAGGGTGGGGCGGATCGC	GACCCCGAACCGCGACCGTAA	150
	U TTATTAGAGGGTGGGGTGGATTGT	CAACCCCAAACCACAACCATAA	151

## Data Availability

Not applicable.

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
