# Peer review of "Helicobacter pylori and Epstein–Barr Virus Co-Infection in Gastric Disease: What Is the Correlation with p53 Mutation, Genes Methylation and Microsatellite Instability in a Cohort of Sicilian Population?"

_ijms, 2023, doi:10.3390/ijms24098104_

Round 1
Reviewer 1 Report
Dear Authors,
why "Gastric Carcinoma" has the acronyms"KG"?please change to GC, Those acronyms "KG" can lead to mistakes.
Studies like this are currently being carried out but in other geographical areas. But, in this case it is focused on a representative population and the authors have highlighted it.
The methodology is correct and the results are expressed in order.
I encourage authors to extend this type of work across the country to avoid bias.
Author Response
Dear Editor,
I would like to thank you for the opportunity of editing the manuscript for IJMS.
My co-authors and I apologize being previously unclear and we hope that the changes we made meet the reviewers’ demands.
In what follows we provide the list of the changes we made (responses are marked as “Reply” while the changes are interline in yellow in the text)
REVIEWER 1
- why "Gastric Carcinoma" has the acronyms"KG"?please change to GC, Those acronyms "KG" can lead to mistakes.
[Reply]: Thank you for your suggests. We changed
- Studies like this are currently being carried out but in other geographical areas. But, in this case it is focused on a representative population and the authors have highlighted it.
[Reply]: OK, Thanks
- The methodology is correct and the results are expressed in order.
[Reply]: OK, Thanks
- I encourage authors to extend this type of work across the country to avoid bias.
[Reply]: Thank you for your comment. A future article will be a multicenter study involving different regions of Italy
Reviewer 2 Report
The topic is actual. H Pylori and EBV infections are widely spread and have a lot of proven implications in gastric cancer.
The title is much too generous regarding the "Italian population" and the number of patients included in the study is only 100.
The study's main limitation, as admitted by the authors, is the small number of patients involved. There are groups with 11 and 18 patients infected with EBV respectively H Pylori. I am sure that statistical analysis is not strong enough.
Also, you must check the spelling, there are a lot of spaces lost between words throughout the whole document.
Author Response
Dear Editor,
I would like to thank you for the opportunity of editing the manuscript for IJMS.
My co-authors and I apologize being previously unclear and we hope that the changes we made meet the reviewers’ demands.
In what follows we provide the list of the changes we made (responses are marked as “Reply” while the changes are interline in yellow in the text)
REVIEWER 2
The topic is actual. H Pylori and EBV infections are widely spread and have a lot of proven implications in gastric cancer.
- The title is much too generous regarding the "Italian population" and the number of patients included in the study is only 100.
[Reply]: Thank you for your comment. We changed the title in:
Helicobacter pylori and Epstein-Barr Virus Co-Infection in Gastric Disease: What is the Correlation with p53 mutation, Genes Methylation and Microsatellite Instability in a cohort of Sicilian population?
- The study's main limitation, as admitted by the authors, is the small number of patients involved. There are groups with 11 and 18 patients infected with EBV respectively H Pylori. I am sure that statistical analysis is not strong enough.
[Reply]: Thank you for your comment. About the statistical analysis including HP (18 pzt) and EBV(11 pzt) group, this analysis was reported in Table 4. In Table 4 we performed a multicomparison tests among patients without infection(n=45), with Hp infection only(n=18), with EBV infection only (n=11) and patients with both Hp and EBV infection(n=26) about all considered parameters. For the statistical analysis among group, we considered
- For continuous data such as age, we perform a test for normal distribution using Shapiro-Wilk test. This test was adeguate for small sample according to Royston P.
- Royston P (1993a) A pocket-calculator algorithm for the Shapiro-Francia test for non-normality: an application to medicine. Statistics in Medicine 12: 181-184.
- Royston P (1993b) A Toolkit for Testing for Non-Normality in Complete and Censored Samples. Journal of the Royal Statistical Society. Series D (The Statistician) 42: 37-43.
- Royston P (1995) A Remark on Algorithm AS 181: The W-test for Normality. Journal of the Royal Statistical Society. Series C (Applied Statistics) 44: 547-551.
- About Kruskal-Wallis test, it is a non-parametric method used for comparing three or more independent samples of equal or different sample sizes. This test provides to following assumptions
- Ordinal Variables– the variable in question should be ordinal or continuous i.e. have some kind of hierarchy to them
- Independence– each group should be independent from the others
- Sample size– each group must have a sample size of 5 or more.
About the use and applicability of Kruskal-Wallis test , we used it according to David J. Sheskin (2003)
Handbook of PARAMETRIC and NONPARAMETRIC STATISTICAL PROCEDURES THIRD EDITION David J. Sheskin Western Connecticut State University(2003)
- For categorial data the multiple comparison tests were performed by chi-square test or Fisher’s exact test. Particularly, Fisher’s exact test was used for small sample sizes and in the case of expected frequencies less than 5, according to David J. Sheskin (2003) Handbook of PARAMETRIC and NONPARAMETRIC STATISTICAL PROCEDURES THIRD EDITION David J. Sheskin Western Connecticut State University(2003)
- Standardized Residuals analysis was performed to individualize the most frequent modality using Z-test with continuity p-value corrected for small samples.
- Finally, chi-square goodness of fit was performed according to following conditions proposed by Koehler and Larnz (1980):
- total of observed counts (N) >= 10
- number of classes (c) >= 3
- all expected values >= 0.25
Also, you must check the spelling, there are a lot of spaces lost between words throughout the whole document.
[Reply]: Done
Reviewer 3 Report
The paper by Anna Giammanco and colleagues reports the results about the molecular characterization of gastric samples from patients with gastric carcinoma, MALT lymphoma, active chronic gastritis or any of these (25 patients in each group).
The results are interesting even if the number of patients in each group is very law to draw robust conclusions.
However, the manuscript need major revisions.
An extensive language revision made by an English native speaker is required, since there are many errors/typos (e.g. lacking spaces etc.).
Materials and Methods - Patients and sample collection: The authors should describe how they selected the 100 patients and the number of patients for each group (normal, gastritis, carcinoma, lymphoma).
The age reported is the age at sampling? Please, clarify it in the text and in table 3.
Please, add information about the tissue from which the DNA has been extracted (normal/tumor/preneoplastic lesion?) especially for the patients without cancer.
Please, add information about the patients with cancer, in particular the stage and the histotype that are linked to the molecular characteristics. These aspects should be also described in detail in the Introduction. E.g. early gastric cancer has peculiar molecular characteristics compared to advanced stages (see PMID: 31763684 and PMID: 33156452). Diffuse type and intestinal type gastric cancer (the second linked to Hp infection) have also different molecular characteristics as demonstrated by the TCGA (PMID: 25079317). The TCGA also showed that methylation is characteristic of 2 gastric cancer subtypes (EBV and MSI) in different genes. CDH1 mutation and/or methylation is a common phenomenon in diffuse gastric cancer (more linked to genetic factors and less to environmental factors) but some authors reported that alterations in the methylation pattern are present also in the intestinal histotype and that Hp is linked to CDH1 methylation (PMID: 34066170 and PMID: 20520968).
The list of the groups with/without Hp/EBV infection should be put in the results and not in the Materials and Methods.
Gene names should always be in italics.
Please check the number of all the tables because Table 1 and 2 are reported in the text as Table 2 and 3, respectively.
Table 1: please write the exon number corresponding to the primers.
The acronyms should be reported in the text the first time (even if they were reported in the abstract). Gastric carcinoma/cancer is usually put in the acronym “GC” instead of “KG”.
Author Response
Dear Editor,
I would like to thank you for the opportunity of editing the manuscript for IJMS.
My co-authors and I apologize being previously unclear and we hope that the changes we made meet the reviewers’ demands.
In what follows we provide the list of the changes we made (responses are marked as “Reply” while the changes are interline in yellow in the text)
REVIEWER 3
An extensive language revision made by an English native speaker is required, since there are many errors/typos (e.g. lacking spaces etc.).
[Reply]: The certificate of revision of english language was added.
Materials and Methods - Patients and sample collection: The authors should describe how they selected the 100 patients and the number of patients for each group (normal, gastritis, carcinoma, lymphoma).
[Reply]: Thank you for your question. We detailed the procedure to define the four group at line:125-130 pag 3
The groups that we have defined in this study were described considering the type of infection starting from a sample composed by 100 patients. These patients were composed by: 25 patients with normal mucosa and patients with 3 groups characterized by types of symptoms that we had identified: active chronic gastritis (n=25), patients with gastric cancer (n=25) and patients with MALT lymphoma (n=25). In this case we chose all groups with same size to reduce possible statistical bias in the definition of the groups with infection
The age reported is the age at sampling? Please, clarify it in the text and in table 3.
[Reply]: Done
Please, add information about the tissue from which the DNA has been extracted (normal/tumor/preneoplastic lesion?) especially for the patients without cancer.
[Reply]: The sentence was added. Lines 123-124
Please, add information about the patients with cancer, in particular the stage and the histotype that are linked to the molecular characteristics.
[Reply]: Done. Supplementary file was added.
These aspects should be also described in detail in the Introduction. E.g. early gastric cancer has peculiar molecular characteristics compared to advanced stages (see PMID: 31763684 and PMID: 33156452). Diffuse type and intestinal type gastric cancer (the second linked to Hp infection) have also different molecular characteristics as demonstrated by the TCGA (PMID: 25079317). The TCGA also showed that methylation is characteristic of 2 gastric cancer subtypes (EBV and MSI) in different genes. CDH1 mutation and/or methylation is a common phenomenon in diffuse gastric cancer (more linked to genetic factors and less to environmental factors) but some authors reported that alterations in the methylation pattern are present also in the intestinal histotype and that Hp is linked to CDH1 methylation (PMID: 34066170 and PMID: 20520968).
[Reply]: In addition as suggested the sentences were added in the text. Lines 103-112
The list of the groups with/without Hp/EBV infection should be put in the results and not in the Materials and Methods.
[Reply]: Done, Line 220-232
Gene names should always be in italics.
[Reply]: Done
Please check the number of all the tables because Table 1 and 2 are reported in the text as Table 2 and 3, respectively.
[Reply]: Done
Table 1: please write the exon number corresponding to the primers.
[Reply]: Done, a column has been inserted in table n°1.
The acronyms should be reported in the text the first time (even if they were reported in the abstract). Gastric carcinoma/cancer is usually put in the acronym “GC” instead of “KG”.
[Reply]: Thank you for your suggests. We changed
Round 2
Reviewer 2 Report
I consider the manuscript improved. I hope that you will extend the research to more subjects.
Reviewer 3 Report
The authors answered all questions, consequently I think that now the manuscript is suitable for publication.